# A Study to Suggest Monthly Baseflow Estimation Approach for the Long-Term Hydrologic Impact Analysis Models: A Case Study in South Korea

**Hanyong Lee, Hyun-Seok Choi, Min-Suh Chae and Youn-Shik Park \***

Department of Rural Construction Engineering, Kongju National University, 54 Daehak-ro, Yesan 32439, Korea; hylee@smail.kongju.ac.kr (H.L.); hschoi@smail.kongju.ac.kr (H.-S.C.); cswoo6432@smail.kongju.ac.kr (M.-S.C.)
\* Correspondence: parkyounshik@gmail.com; Tel.: +82-41-330-1267

**Abstract:** Changes in both land use and rainfall patterns can lead to changes in the hydrologic behavior of the watershed. The long-term hydrologic impact analysis (L-THIA) model has been used to predict such changes and analyze the changes in mitigation scenarios. The model is simple as only a small amount of input data are required, but it can predict only the direct runoff and cannot determine the streamflow. This study, therefore, aimed to propose a method for predicting the monthly baseflow while maintaining the simplicity of the model. The monthly baseflows for 20 watersheds in South Korea were estimated under different land use conditions. Calibration of the monthly baseflow prediction method produced values for $R^2$ and the Nash–Sutcliffe efficiency (NSE) within the ranges of 0.600–0.817 and 0.504–0.677, respectively; during validation, these values were in the ranges of 0.618–0.786 and 0.567–0.727, respectively. This indicates that the proposed method can reliably predict the monthly baseflow while maintaining the simplicity of the L-THIA model. The proposed model is expected to be applicable to all the various forms of the model.

**Keywords:** L-THIA model; monthly baseflow; hydrological watershed modeling

## 1. Introduction

Changes in land use or rainfall pattern not only change the behavior of direct runoff and baseflow in the watershed, but also affect the occurrence of nonpoint source (NPS) pollution. Urbanization and industrialization have increased the ratio of impervious surfaces in watersheds. Therefore, it is necessary to analyze such changes in land use and investigate scenarios that may reduce the impacts of these changes. Hydrologic models are generally used for such analyses; Bieger et al. [1] assessed the impact of land use changes using the soil and water assessment tool (SWAT) [2,3], and three land use scenarios for forest, cropland, and orchard area changes were established. The result indicated that forest, cropland, and orchard area changes of −34.48%, +265.32%, and 204.51% led to surface flow increases of 46.1%. Guse et al. [4] analyzed the impact of spatially distributed five crop rotations using the SWAT model; nitrate loads were reduced with dynamic changes in agricultural crop rotations. Martin et al. [5] used the regional hydro-ecological simulation system model [6]; the study result indicated that high flows (highest 10 percentiles) increased by 37–88% and that low flows (lowest 10 percentiles) increased by 23–37% by land use changes. Additionally, Srivastava et al. [7] and Aghsaei et al. [8] reported that vegetation can provide a significant effect on hydrological components with the variations in the physical characteristics of the land surface, soil, and vegetation, which are the roughness, albedo, architectural resistance, infiltration capacity, leaf area index, root depth, and stomatal conductance.

The long-term hydrologic impact analysis (L-THIA) model has been in use for this purpose since 1994 [9]. The L-THIA model was first developed in the form of a spreadsheet in 1994 [9], followed by redevelopment so that it could be integrated with geographic

information systems (GIS) [10,11]. However, the model is limited in terms of reflecting the various types of land use. To counter such limitations, Lim et al. [12] developed the L-THIA/NPS WWW model so that L-THIA could be used in the form of a spreadsheet in L-THIA/NPS GIS, which is based on the ArcView software. Bhaduri et al. [13] used the GIS-based L-THIA/NPS GIS model to predict changes in the direct runoff and pollution from nonpoint sources resulting from land use changes; they reported that the average annual direct runoff increased by 80%, while with pollution from lead, copper, and zinc increased by more than 50% as the urban and impervious area increased by 18% during the period 1973–1991. Wilson and Weng [14] analyzed changes in the direct runoff that were caused by land use changes using the L-THIA NPS model, which is an ArcHydro GIS extension. They reported a two-fold increase in the direct runoff in places where the residential land use increased by 37.3%, even though the precipitation increased by <30%. Liu et al. [15] considered best management practices (BMPs) such as the implementation of wet ponds, green roofs, and bioretention or the construction cost, annual maintenance cost, and interest rate of 12 techniques that belong to low impact development (LIDs) using the L-THIA-LID model [16]. They determined cost-effective strategies for 15 scenarios to reduce direct runoff, total nitrogen (TN), total phosphorus (TP), total suspended solids (TSS), Pb, biochemical oxygen demand (BOD), and chemical oxygen demand (COD). Eaton [17] analyzed a direct runoff reduction method using the L-THIA LID model to analyze green infrastructure screening and reported that direct runoff can be reduced by 12% by using bioretention and raingardens in the watershed. Li et al. [18] analyzed changes in the surface runoff caused by land use and rainfall changes using the ArcL-THIA 10.1 model that is based on ArcGIS 10.0 [19]. The authors calibrated the ArcL-THIA 10.1 model by using the baseflow filter program (BFLOW) [20] model to separate direct runoff; they reported that enhanced precipitation contributed more significantly to the observed changes compared to land use during the period 2005–2015.

Different methods for analyzing scenarios resulting from changes in land use or rainfall conditions, which also consider baseflow, have been proposed for the L-THIA model. Ahiablame et al. [21] analyzed the effects of rain barrel/cistern and porous pavement using the LTHIA-LID model in their study, which reflected the influence of baseflow. The annual baseflow regression equation, which includes the area of the watershed, annual precipitation, and baseflow index, was utilized for baseflow analysis in the L-THIA model. However, as this method can only predict the annual baseflow, it has limitations when used to determine monthly characteristics. Ryu et al. [22] improved the prediction process of the existing L-THIA in detail by using the asymptotic curve number (ACN) instead of CN for predicting the direct runoff. This made it possible to predict streamflow by adding modules for baseflow predictions and channel routing to overcome the most significant limitations of the existing L-THIA. However, this process involves an increase in the complexity of the model as three additional model parameters related to direct runoff prediction, four model parameters related to baseflow prediction, and three model parameters related to channel routing prediction were required, and the need for an optimization algorithm was suggested for calibrating the model.

The L-THIA model is based on spreadsheets [9,16] and GIS [10–14,19]. It has been in continuous use since its conception for analyzing direct runoff resulting from changes in land use or rainfall patterns as well as the effects of BMP and LID techniques. However, a comparison of the predictions made using this model with actual streamflow is required to improve the utility of this model. In addition, it is necessary to reflect the influence of baseflow on the analysis process based on this model. However, the model is generally used for the above purposes because of its simplicity, as it requires only input data to define CN and its computation process is not complicated. It is, therefore, necessary to maintain the current benefits of this model while predicting baseflow. This study aims to propose a method for predicting the baseflow while maintaining the simplicity of the current L-THIA model.

## 2. Materials and Methods

### 2.1. Decription of the Study Area

As the purpose of this study is the proposal of a method that can predict baseflow while maintaining the simplicity of the current L-THIA model, streamflow data describing the runoff in a watershed, the land use map, and monthly precipitation data were required as input into the L-THIA model.

The land use map (scale at 1:5000) was provided by the Environmental Geographic Information Service [23]; the monthly precipitation data were provided by the Korea Meteorological Administration [24]; the daily stream flow data were acquired from the Water Resources Management Information System (WAMIS) [25]. These data were used in this study. The watersheds selected for the study include the flow data measurement points operated by WAMIS as watershed outlets, and they are distributed such that the conditions in various regions of Korea are considered without any spatial overlap (Figure 1). A total of 20 watersheds were selected with areas ranging from 5694.8 to 155,805.9 ha (Table 1). The land use types were classified into urban, agriculture, forest, pasture, wetland, bare land, and water. Forest occupied the largest area in all the watersheds studied, followed by agriculture. The exception to this trend was Wsd-02 in which agriculture (2741.0 ha) represented the largest proportion (41.5%), followed by urban (2042.1 ha), which accounted for 30.9% of the watershed area (Table 1).

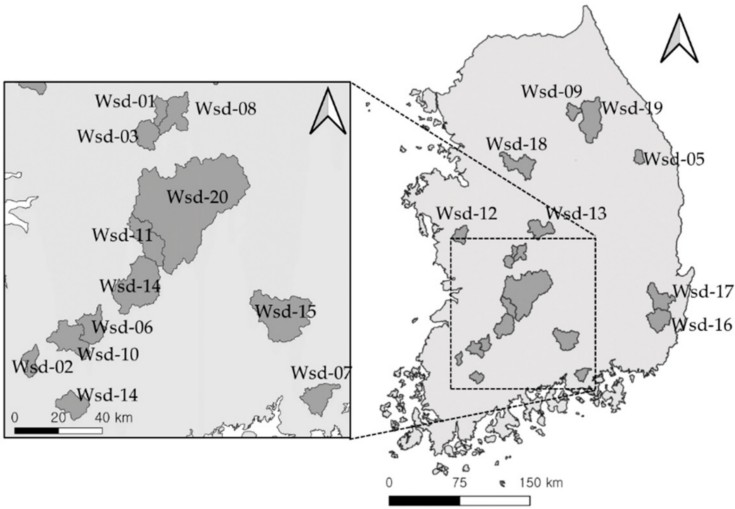

**Figure 1.** Location of study watershed.

**Table 1.** Land uses in the study watersheds.

| Watershed | Area (ha) | | | | | | | |
|---|---|---|---|---|---|---|---|---|
| | Urban | Agriculture | Forest | Pasture | Wetland | Bare land | Water | Total |
| Wsd-01 | 590.3 | 372.2 | 4637.7 | 27.1 | 26.1 | 41.3 | 0.1 | 5694.8 |
| Wsd-02 | 2042.1 | 2741.0 | 1447.9 | 199.0 | 36.1 | 121.7 | 12.2 | 6600.0 |
| Wsd-03 | 196.3 | 1169.7 | 9256.9 | 82.7 | 9.1 | 121.8 | 9.4 | 10,845.8 |
| Wsd-04 | 569.7 | 1536.9 | 9425.8 | 142.7 | 63.4 | 126.7 | 47.8 | 11,912.9 |
| Wsd-05 | 656.4 | 479.5 | 11,134.4 | 214.9 | 0.0 | 142.8 | 2.3 | 12,630.2 |

**Table 1.** *Cont.*

| Watershed | Area (ha) | | | | | | | |
|---|---|---|---|---|---|---|---|---|
| | Urban | Agriculture | Forest | Pasture | Wetland | Bare land | Water | Total |
| Wsd-06 | 435.7 | 3361.2 | 9002.8 | 224.6 | 4.9 | 155.4 | 36.0 | 13,220.6 |
| Wsd-07 | 131.4 | 1979.7 | 11,115.2 | 281.0 | 13.1 | 72.2 | 108.5 | 13,701.0 |
| Wsd-08 | 983.1 | 2266.7 | 11,494.5 | 565.4 | 35.9 | 149.0 | 45.5 | 15,540.0 |
| Wsd-09 | 20.2 | 1472.9 | 15,794.5 | 92.6 | 0.0 | 18.0 | 0.0 | 17,398.2 |
| Wsd-10 | 843.6 | 5017.5 | 11,771.9 | 269.2 | 112.4 | 424.5 | 131.6 | 18,570.7 |
| Wsd-11 | 165.2 | 2317.5 | 16,250.0 | 239.0 | 0.3 | 55.8 | 76.1 | 19,103.8 |
| Wsd-12 | 343.8 | 5118.2 | 14,822.7 | 200.6 | 85.8 | 152.5 | 111.8 | 20,835.4 |
| Wsd-13 | 770.9 | 5908.7 | 26,938.4 | 934.1 | 288.9 | 289.4 | 301.8 | 35,432.1 |
| Wsd-14 | 404.8 | 7297.3 | 26,403.6 | 1158.6 | 0.1 | 384.5 | 179.5 | 35,828.3 |
| Wsd-15 | 278.2 | 4779.9 | 35,251.6 | 327.5 | 120.3 | 228.2 | 342.6 | 41,328.3 |
| Wsd-16 | 2454.4 | 5312.6 | 32,486.2 | 1044.5 | 186.9 | 655.1 | 618.9 | 42,758.7 |
| Wsd-17 | 1347.9 | 8703.7 | 31,043.6 | 1322.7 | 259.1 | 320.0 | 380.6 | 43,377.7 |
| Wsd-18 | 2098.6 | 21,737.3 | 23,612.5 | 2642.9 | 49.7 | 1319.6 | 480.9 | 51,941.4 |
| Wsd-19 | 347.6 | 5614.7 | 73,684.2 | 601.7 | 0.1 | 371.1 | 94.0 | 80,713.3 |
| Wsd-20 | 1763.3 | 17,163.2 | 130,735.8 | 1857.5 | 274.1 | 644.0 | 3368.2 | 155,805.9 |

The analysis period was set to the five-year period from 1 January 2016 to 31 December 2020. The minimum monthly precipitation in each watershed ranged from 0.5 mm (Wsd-01) to 8.1 mm (Wsd-07), whereas the maximum monthly precipitation ranged from 454.5 mm (Wsd-17) to 822.0 mm (Wsd-14). The maximum precipitation was, therefore, 73 times (Wsd-07) to 1271 times (Wsd-01) higher than the minimum in each watershed, indicating a significant difference in monthly precipitation (Table 2).

**Table 2.** Statistics of monthly precipitation and daily stream flow.

| Watershed | Monthly Precipitation (mm) | | Daily Stream Flow (m³/s) | | |
|---|---|---|---|---|---|
| | min. | max. | min. | max. | Mean |
| Wsd-01 | 0.5 | 635.5 | 0.02 | 118.65 | 1.31 |
| Wsd-02 | 2.1 | 738.1 | 0.16 | 341.68 | 1.98 |
| Wsd-03 | 4.1 | 628.0 | 0.01 | 241.93 | 3.20 |
| Wsd-04 | 2.1 | 738.1 | 0.01 | 191.06 | 2.71 |
| Wsd-05 | 1.5 | 494.4 | 0.07 | 133.19 | 3.18 |
| Wsd-06 | 3.5 | 822.0 | 0.01 | 1496.45 | 4.33 |
| Wsd-07 | 8.1 | 587.5 | 0.01 | 247.78 | 4.03 |
| Wsd-08 | 2.0 | 469.5 | 0.13 | 380.58 | 4.07 |
| Wsd-09 | 1.0 | 761.5 | 0.10 | 189.60 | 3.13 |
| Wsd-10 | 2.1 | 738.1 | 0.13 | 995.01 | 5.38 |
| Wsd-11 | 7.6 | 631.4 | 0.02 | 618.54 | 5.49 |
| Wsd-12 | 0.5 | 492.6 | 0.05 | 202.72 | 4.06 |
| Wsd-13 | 0.9 | 588.1 | 0.05 | 312.41 | 6.78 |
| Wsd-14 | 3.5 | 822.0 | 0.01 | 661.89 | 7.53 |
| Wsd-15 | 0.8 | 712.0 | 0.10 | 894.30 | 10.44 |
| Wsd-16 | 5.7 | 498.3 | 0.30 | 697.60 | 10.28 |
| Wsd-17 | 1.8 | 451.5 | 0.01 | 702.76 | 7.73 |
| Wsd-18 | 1.3 | 616.7 | 0.13 | 1039.87 | 9.39 |
| Wsd-19 | 0.7 | 606.5 | 0.60 | 736.20 | 17.99 |
| Wsd-20 | 4.1 | 628.0 | 0.50 | 1160.00 | 21.72 |

The minimum daily stream flow ranged from 0.01 $m^3$/s (Wsd-03, Wsd-04, Wsd-06, Wsd-07, Wsd-14, and Wsd-17) to 0.60 $m^3$/s (Wsd-19), whereas the maximum daily stream flow ranged from 118.65 $m^3$/s (Wsd-01) to 1160.00 $m^3$/s (Wsd-20). The average daily stream flow ranged from 1.31 $m^3$/s (Wsd-01) to 21.72 $m^3$/s (Wsd-20) (Table 2). The difference between the minimum and maximum daily stream flow was significant in each

watershed. Therefore, it was judged necessary to reflect monthly precipitation conditions in predicting the monthly baseflow.

### 2.2. Baseflow Separation

As the purpose of this study is to propose a method for predicting monthly baseflow, measurement of the actual baseflow is required for comparison. The measured streamflow is the sum of the surface runoff and the baseflow, which means that it is difficult to evaluate the monthly baseflow predictions method solely by comparison with the corresponding streamflow. Therefore, the monthly baseflow prediction method was examined by separating the baseflow from the streamflow in each watershed using the method suggested by Eckhardt (Equation (1)). In this baseflow separation method [26], the baseflow on day $t$ ($b_t$) is determined using the maximum value of the long-term ratio of baseflow to streamflow ($BFI_{max}$), the filter parameter ($a$), the filtered baseflow at the $t-1$ time step ($b_{t-1}$), and the streamflow at t time step ($Q_t$):

$$b_t = \frac{(1 - BFI_{max})ab_{t-1} + (1-a)BFI_{max}Q_t}{1 - aBFI_{max}} \tag{1}$$

After separating the baseflow ($m^3/s$) from the streamflow ($m^3/s$) in each watershed using Equation (1), the baseflow can be expressed (in $m^3$). It reflects the area of each watershed as shown in Table 3. Overall, the minimum, maximum, and average values of the streamflow and baseflow showed a tendency to rise as the area of the watershed increased. The mean flow percentage (%), which is the ratio of baseflow to streamflow, ranged from 32.7% (Wsd-15) to 59.2% (Wsd-19), indicating significant differences among the mean flow percentages in each watershed despite the seemingly insignificant differences observed using the actual numbers. Wsd-15, with a relatively large watershed area, exhibited the minimum percentage of 32.7%; however, the mean flow percentage showed a tendency to slightly increase as the watershed area increased. This indicates that it is necessary to consider parameters related to the area under investigation when predicting the monthly baseflow.

**Table 3.** Statistics describing the streamflow and separated baseflow.

| Watershed | Streamflow ($\times 10^6$ m$^3$) | | | Baseflow ($\times 10^6$ m$^3$) | | | Mean Flow Percentage (%) |
|---|---|---|---|---|---|---|---|
| | min. | max. | Mean | min. | max. | Mean | |
| Wsd-01 | 0.136 | 28.600 | 3.544 | 0.104 | 8.862 | 1.552 | 43.8 |
| Wsd-02 | 1.045 | 44.823 | 5.211 | 0.763 | 11.033 | 2.240 | 43.0 |
| Wsd-03 | 0.003 | 69.717 | 8.160 | 0.002 | 20.880 | 3.520 | 43.1 |
| Wsd-04 | 0.185 | 57.876 | 7.011 | 0.037 | 20.724 | 3.010 | 42.9 |
| Wsd-05 | 0.710 | 41.993 | 8.360 | 0.428 | 15.645 | 4.026 | 48.2 |
| Wsd-06 | 0.219 | 214.748 | 11.375 | 0.178 | 59.278 | 4.752 | 41.8 |
| Wsd-07 | 0.535 | 66.047 | 10.290 | 0.314 | 24.568 | 4.378 | 42.6 |
| Wsd-08 | 0.743 | 98.042 | 10.432 | 0.491 | 27.205 | 4.254 | 40.8 |
| Wsd-09 | 0.164 | 76.421 | 8.166 | 0.046 | 26.349 | 3.792 | 46.4 |
| Wsd-10 | 2.035 | 172.189 | 14.231 | 1.065 | 61.150 | 6.888 | 48.4 |
| Wsd-11 | 0.328 | 147.041 | 14.450 | 0.236 | 54.088 | 6.187 | 42.8 |
| Wsd-12 | 1.457 | 75.855 | 11.120 | 1.168 | 38.142 | 5.956 | 53.6 |
| Wsd-13 | 1.456 | 110.265 | 18.222 | 0.632 | 61.698 | 9.312 | 51.1 |
| Wsd-14 | 0.902 | 151.183 | 19.463 | 0.178 | 66.078 | 10.207 | 52.4 |
| Wsd-15 | 0.130 | 199.930 | 26.542 | 0.052 | 78.220 | 8.681 | 32.7 |
| Wsd-16 | 3.707 | 117.337 | 27.034 | 1.972 | 44.484 | 13.595 | 50.3 |
| Wsd-17 | 0.212 | 95.168 | 19.108 | 0.041 | 33.397 | 7.194 | 37.6 |
| Wsd-18 | 0.481 | 280.725 | 24.998 | 0.421 | 123.584 | 13.505 | 54.0 |
| Wsd-19 | 7.379 | 375.978 | 48.052 | 4.087 | 194.168 | 28.456 | 59.2 |
| Wsd-20 | 8.027 | 524.906 | 57.826 | 5.825 | 201.185 | 30.414 | 52.6 |

*2.3. Monthly Baseflow Estimation Approach*

A remarkable method that can be used for baseflow prediction is the spreadsheet tool for the estimation of pollutant load (STEPL) [27] model. This model was proposed by the U. S. Environmental Protection Agency (U. S. EPA) to establish a total maximum daily load plan for the United States, and it can analyze the effects of more than 50 BMPs while also simulating the average annual runoff, sediment, nitrogen, phosphorus, and the BOD. In this model, prediction of the baseline is based on the annual precipitation, land use, and hydrologic soil group (HSG). For example, it predicts that the baseflow will correspond to 36, 24, 12, and 6% of the annual precipitation in an urban area when the HSG is A, B, C, and D, respectively. In this model, land uses other than urban include cropland, pasture, and forest, but the baseflow is estimated using the same method for all types of area. In other words, the model predicts that the baseflow will correspond to 45, 30, 15, and 7.5% of the annual precipitation for the HSGs A, B, C, and D, respectively. Since this method is a statistical approach with a simple prediction process, it was judged that the application of this method could also maintain model simplicity in the baseflow prediction process of the L-THIA model.

A basic formula is required to predict the monthly baseflow. Land use conditions need to be reflected because the baseflow is likely to vary alongside land use. It was also judged that monthly rainfall requires consideration because both the baseflow and streamflow are affected by precipitation. The basic formula for predicting the monthly baseflow is shown in Equation (2). For predicting the baseflow in month $i$, this equation considers the conditions under which precipitation occurs in month $i$, the coefficient for urban land ($C_{URBN}$) and the area covered by urban land ($A_{URBN}$), the coefficient for agriculture ($C_{AGRL}$) and the area of agricultural land ($A_{AGRL}$), the coefficient for forest ($C_{FRST}$) and the area of forested land ($A_{FRST}$), the coefficient for pasture ($C_{PAST}$) and the area covered by pasture ($A_{PAST}$), the coefficient for wetland ($C_{WTLD}$) and the area covered by wetland ($A_{WTLD}$), the coefficient for bare land ($C_{BARE}$) and the area covered by bare land ($A_{BARE}$), and the coefficient for water ($C_{WATR}$) and the area covered by water ($A_{WATR}$).

$$
\begin{aligned}
Baseflow_i = Precipitation_i \times (&C_{URBN} \times A_{URBN} + C_{AGRL} \times A_{AGRL} \\
&+C_{FRST} \times A_{FRST} + C_{PAST} \times A_{PAST} + C_{WTLD} \times A_{WTLD} \\
&+C_{BARE} \times A_{BARE} + C_{WATR} \times A_{WATR})
\end{aligned} \tag{2}
$$

Equation (2) can reflect the conditions surrounding both monthly precipitation and land use for predicting the baseflow, as the coefficients allow the degree of influence from each land use type to vary. However, the coefficients require definition, for which a genetic algorithm (GA) [28] was used in this case. GA, which is similar to the evolutionary process of nature, is used to obtain the optimized solution to a given problem. It is, therefore, useful in solving highly complex problems in the fields of business or engineering [29–31]. In this study, the selected watersheds were divided into two groups for determination of the seven coefficients. The first group included 11 watersheds. It consisted of the watersheds that were given even numbers: Wsd-01, Wsd-02, Wsd-04, Wsd-06, Wsd-08, Wsd-10, Wsd-12, Wsd-14, Wsd-16, Wsd-18, and Wsd-20, and it included the watersheds with the minimum and maximum areas. The second watershed group consisted of the remaining nine watersheds: Wsd-03, Wsd-05, Wsd-07, Wsd-09, Wsd-11, Wsd-13, Wsd-15, Wsd-17, and Wsd-19. After the coefficients of Equation (2) were defined through GA for the watersheds in the first group, Equation (2) was applied to the watersheds in the second group, along with a definition for all the coefficients, to examine the monthly baseflow prediction method. The processes described in Section 2.1 can be expressed as the following Figure 2.

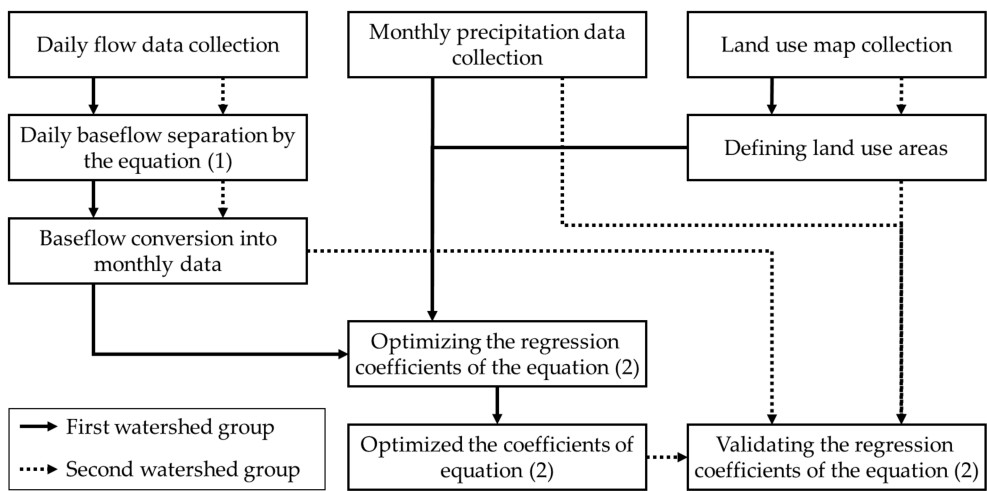

**Figure 2.** Schematic flow of the study.

## 3. Results and Discussion

### 3.1. Determination of Regression Model Coefficients

Optimal values for the seven coefficients in Equation (2) for predicting the monthly baseflow were determined by GA based on the monthly baseflow estimated for each watershed and the monthly flow that was distinguished using the Eckhardt filter equation (Equation (1)). The coefficients for which the smallest difference between the monthly baseflow separated using Equation (1) and the monthly baseflow estimated using Equation (2) was observed in each watershed were determined to be optimal. In this study, it was deemed necessary to measure the separated monthly baseflows and to develop the criteria for examining the validity of the estimated monthly baseflow. Duda et al. [32] reported that the estimated result is applicable when the $R^2$ is higher than 0.65 and the difference is 45% or less. Skaggs et al. [33] mentioned that the result is applicable when the Nash–Sutcliffe efficiency (NSE) is higher than 0.50, and Wang et al. [34] suggested that it is applicable when the NSE is higher than 0.50, the $R^2$ is higher than 0.60, and the PBIAS is ±15%. Moriasi et al. [35] deemed the result to be applicable when the $R^2$ is higher than 0.60, NSE is higher than 0.50, and the PBIAS is less than 15%. In other words, there are various criteria for determining the applicability of the estimated result. The different criteria were summarized in this study, and the applicability of the monthly baseflow was estimated using a scatter plot with an NSE higher than 0.50 and an $R^2$ higher than 0.60.

In terms of the optimized coefficients for the 11 watersheds in the first group, $C_{URBN}$ ranged from 0.00289 (Wsd-04) to 0.06314 (Wsd-04), $C_{AGRL}$ from 0.10076 (Wsd-01) to 0.49061 (Wsd-06), $C_{FRST}$ from 0.01256 (Wsd-02) to 0.28413 (Wsd-10), $C_{PAST}$ from 0.0.07913 (Wsd-06) to 0.31460 (Wsd-20), $C_{WTLD}$ from 0.05222 (Wsd-02) to 0.78599 (Wsd-16), $C_{BARE}$ from 0.03488 (Wsd-04) to 0.23603 (Wsd-16), and $C_{WATR}$ from 0.04919 (Wsd-04) to 0.38384 (Wsd-16) (Table 4). The optimized coefficients in each watershed were obtained when the difference between the separated and estimated baseflows describing baseflow was minimal. Since the purpose of this study is to propose a method for predicting the monthly baseflow in multiple watersheds rather than judging the accuracy of the monthly baseflow prediction for a specific watershed, the final values of the coefficients of Equation (2) were determined based on the average of each coefficient. The coefficients were, therefore, determined to be 0.04 for $C_{URBN}$, 0.40 for $C_{AGRL}$, 0.20 for $C_{FRST}$, 0.18 for $C_{PAST}$, 0.48 for $C_{WTLD}$, 0.15 for $C_{BARE}$, and 0.22 for $C_{WATR}$.

**Table 4.** Optimized coefficients for the Equation (2).

| Watershed | $C_{URBN}$ | $C_{AGRL}$ | $C_{FRST}$ | $C_{PAST}$ | $C_{WTLD}$ | $C_{BARE}$ | $C_{WATR}$ |
|---|---|---|---|---|---|---|---|
| Wsd-01 | 0.02241 | 0.10076 | 0.25373 | 0.09274 | 0.52890 | 0.09265 | 0.04994 |
| Wsd-02 | 0.03981 | 0.48080 | 0.01256 | 0.18411 | 0.05222 | 0.21955 | 0.24430 |
| Wsd-04 | 0.06314 | 0.19460 | 0.21289 | 0.24017 | 0.70207 | 0.03488 | 0.04919 |
| Wsd-06 | 0.03137 | 0.49061 | 0.27421 | 0.07913 | 0.24028 | 0.19144 | 0.14365 |
| Wsd-08 | 0.02437 | 0.35966 | 0.26696 | 0.08260 | 0.17413 | 0.09301 | 0.18265 |
| Wsd-10 | 0.05851 | 0.42420 | 0.28413 | 0.23152 | 0.47602 | 0.21825 | 0.31505 |
| Wsd-12 | 0.06050 | 0.23236 | 0.25996 | 0.15333 | 0.67417 | 0.11600 | 0.20427 |
| Wsd-14 | 0.03400 | 0.39035 | 0.14121 | 0.28233 | 0.70807 | 0.17528 | 0.29423 |
| Wsd-16 | 0.00289 | 0.33503 | 0.24279 | 0.12178 | 0.78599 | 0.23603 | 0.38384 |
| Wsd-18 | 0.03494 | 0.34299 | 0.15810 | 0.17155 | 0.27299 | 0.23278 | 0.20131 |
| Wsd-20 | 0.04969 | 0.47828 | 0.12389 | 0.31460 | 0.61544 | 0.04141 | 0.30796 |
| Min. | 0.00289 | 0.10076 | 0.01256 | 0.07913 | 0.05222 | 0.03488 | 0.04919 |
| Max. | 0.06314 | 0.49061 | 0.28413 | 0.31460 | 0.78599 | 0.23603 | 0.38384 |
| Mean | 0.03833 | 0.34815 | 0.20277 | 0.17762 | 0.47548 | 0.15012 | 0.21603 |
| Final value | 0.04 | 0.40 | 0.20 | 0.18 | 0.48 | 0.15 | 0.22 |

In general, the contribution of land use to the baseflow can be considered to be related to the impervious surface ratio. The contribution of urban areas to the baseflow will be low because the impervious surface ratio is high. A $C_{URBN}$ of 0.04 was finally determined for urban areas based on the optimized results; this reflects the conditions of impermeability, as this coefficient is relatively lower than those obtained for other land uses. In contrast, the contribution of wetlands and reservoirs to the baseflow is high because of the constant infiltration of water. The value of 0.48 for $C_{WTLD}$ also appears to reflect this condition, as it is relatively high as compared to the coefficients for other types of land use.

It is noteworthy that the coefficient for agricultural land, $C_{AGRL}$, had the second highest value after $C_{WTLD}$. Agriculture in Korea is dominated by rice paddies, which are maintained in pond conditions during the rice cultivation period from May to October, resulting in a similar contribution to the baseflow as wetland. The contribution of agriculture to the baseflow should, therefore, be similar to that of wetland; this condition is sufficiently reflected in the coefficient for agriculture.

The monthly baseflow in the first watershed group was calculated again using Equation (2) by applying the finally determined coefficients, and the suitability of the monthly baseflow was determined by the values of $R^2$, NSE, and the scatter plot. Both $R^2$ and NSE showed applicable ranges with $R^2$ ranging from 0.600 (Wsd-06) to 0.817 (Wsd-16) and NSE from 0.504 (Wsd-01) to 0.677 (Wsd-18 and Wsd-20) (Table 5). Based on the scatter plot, the estimated monthly baseflow tended to be slightly lower than the separated monthly baseflow for Wsd-06, Wsd-08, and Wsd-10; Wsd-20 showed a scattered tendency that was comparable to the other watersheds. Overall, however, there were no significant differences in the tendencies or values obtained via prediction and the separated monthly baseflow (Figure 3).

**Table 5.** $R^2$ and NSE of separated and estimated monthly baseflow in the first watershed group.

| Watershed | $R^2$ | NSE |
|---|---|---|
| Wsd-01 | 0.736 | 0.504 |
| Wsd-02 | 0.607 | 0.563 |
| Wsd-04 | 0.634 | 0.507 |
| Wsd-06 | 0.600 | 0.565 |
| Wsd-08 | 0.704 | 0.679 |

**Table 5.** *Cont.*

| Watershed | $R^2$ | NSE |
|---|---|---|
| Wsd-10 | 0.708 | 0.536 |
| Wsd-12 | 0.697 | 0.547 |
| Wsd-14 | 0.688 | 0.583 |
| Wsd-16 | 0.817 | 0.623 |
| Wsd-18 | 0.785 | 0.677 |
| Wsd-20 | 0.691 | 0.677 |

In the plot of flow duration curves, the estimated monthly baseflow did not capture the separated monthly baseflow in the dry-conditions (flow duration intervals from 60% to 90%) and the low-flow (flow duration intervals from 90% to 100%) regimes often; however, it does in the other flow regimes, which are the high-flow (flow duration intervals from 0–10%), the moist-conditions (flow duration intervals from 10% to 40%), and the mid-range flow (flow duration intervals from 40% to 60%) regimes (Figure 4).

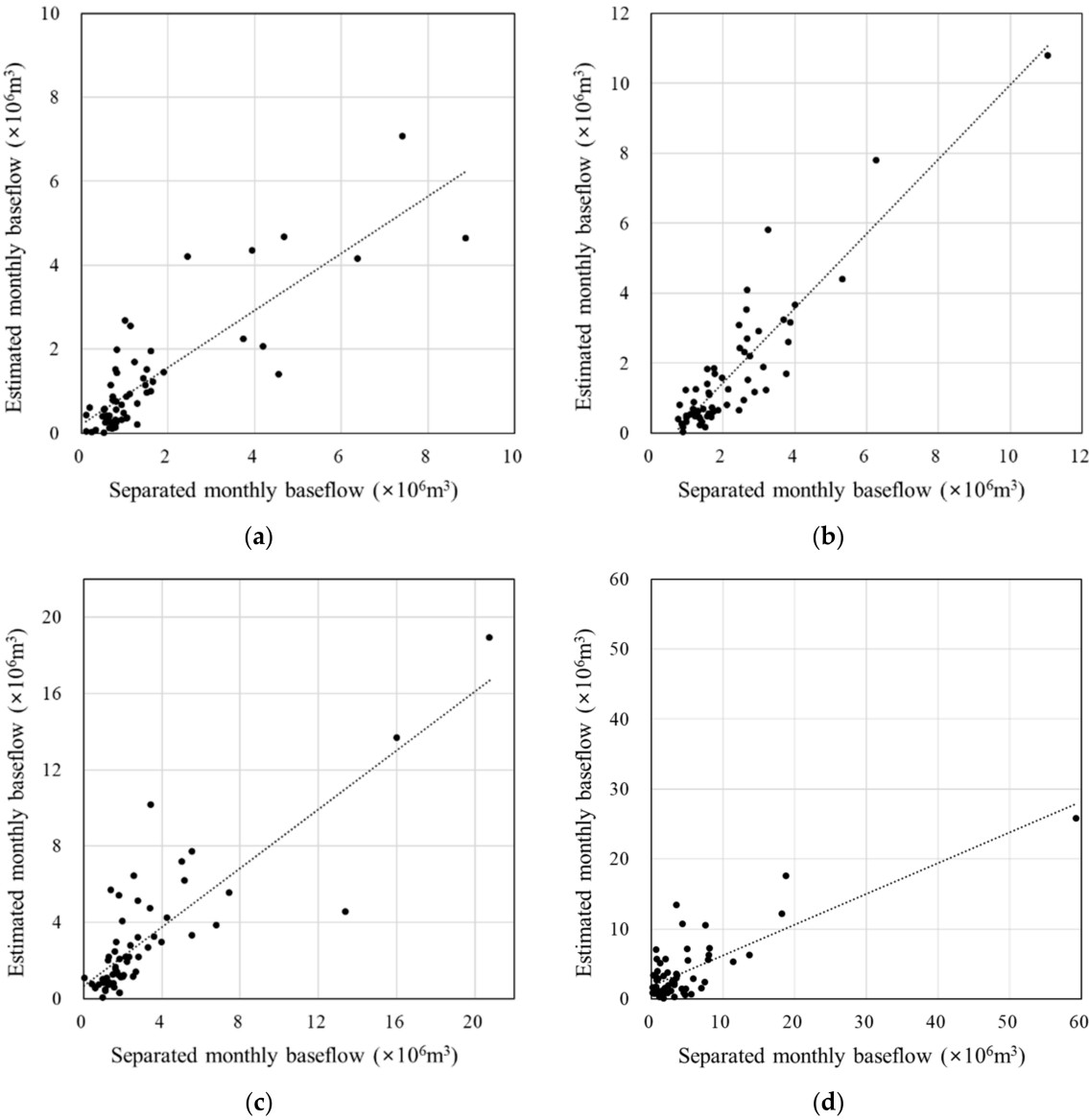

**Figure 3.** *Cont.*

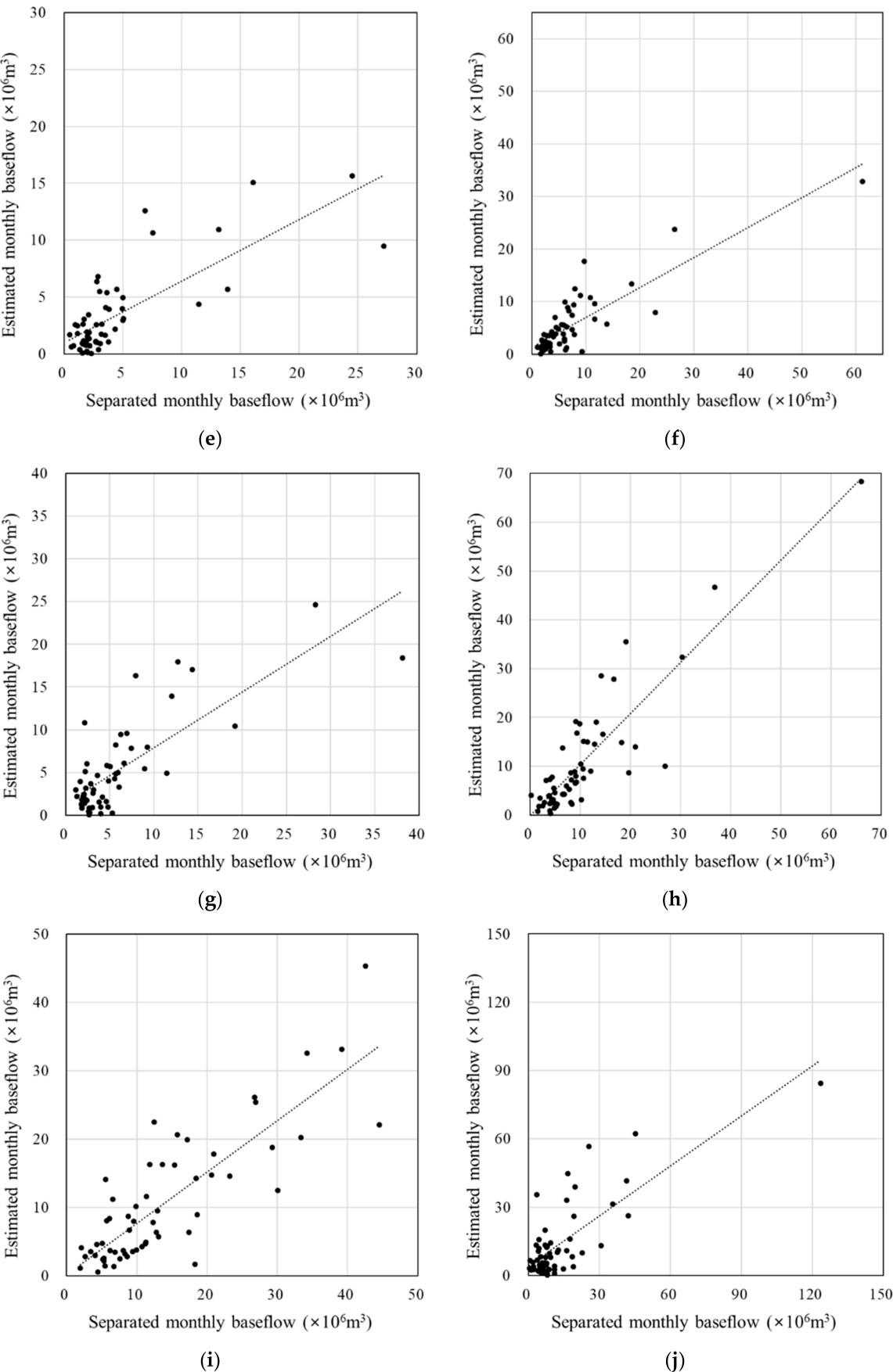

**Figure 3.** *Cont.*

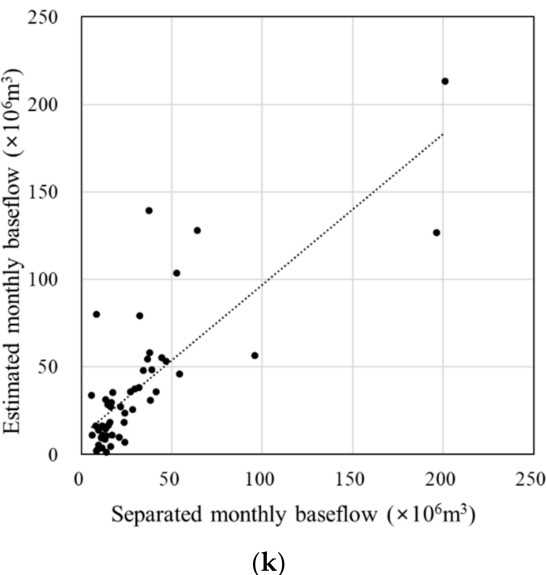

(**k**)

**Figure 3.** Scatter plot of separated monthly baseflow and estimated monthly baseflow: (**a**) Wsd-01; (**b**) Wsd-02; (**c**) Wsd-04; (**d**) Wsd-06; (**e**) Wsd-08; (**f**) Wsd-10; (**g**) Wsd-12; (**h**) Wsd-14; (**i**) Wsd-16; (**j**) Wsd-18; (**k**) Wsd-20.

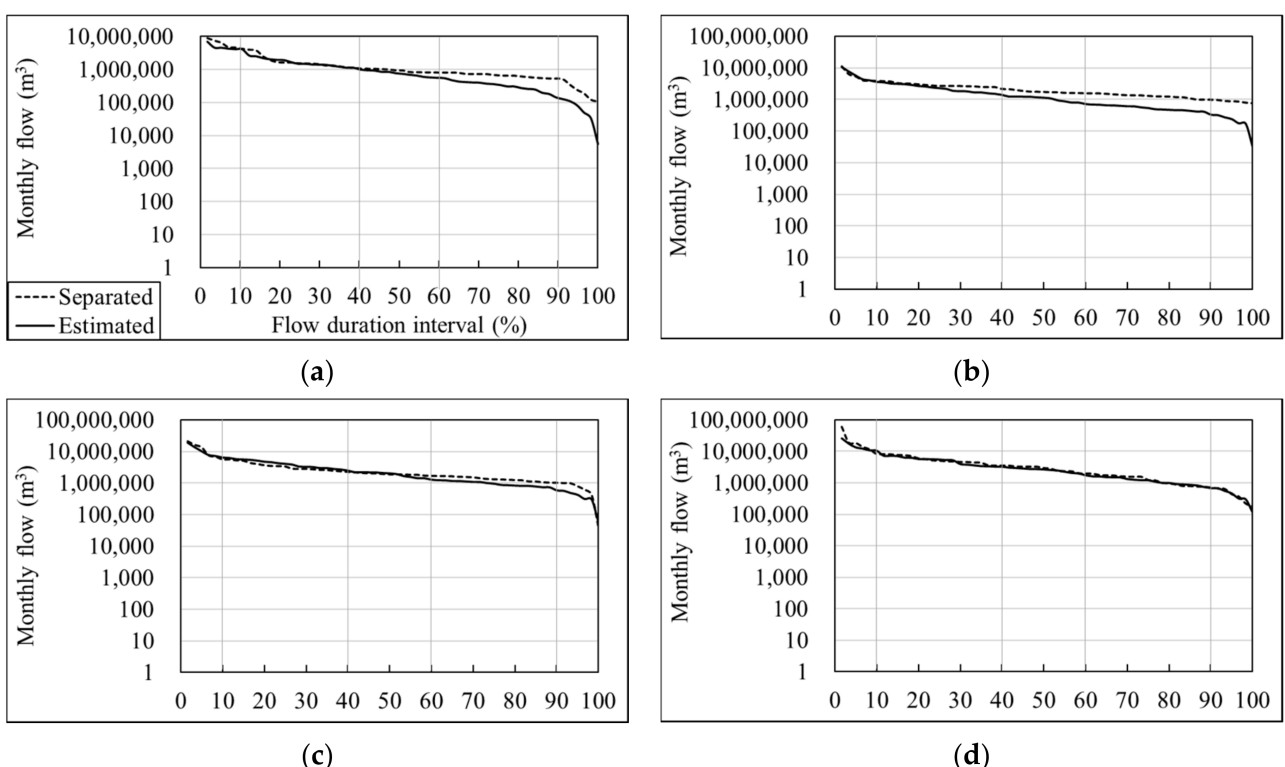

**Figure 4.** *Cont.*

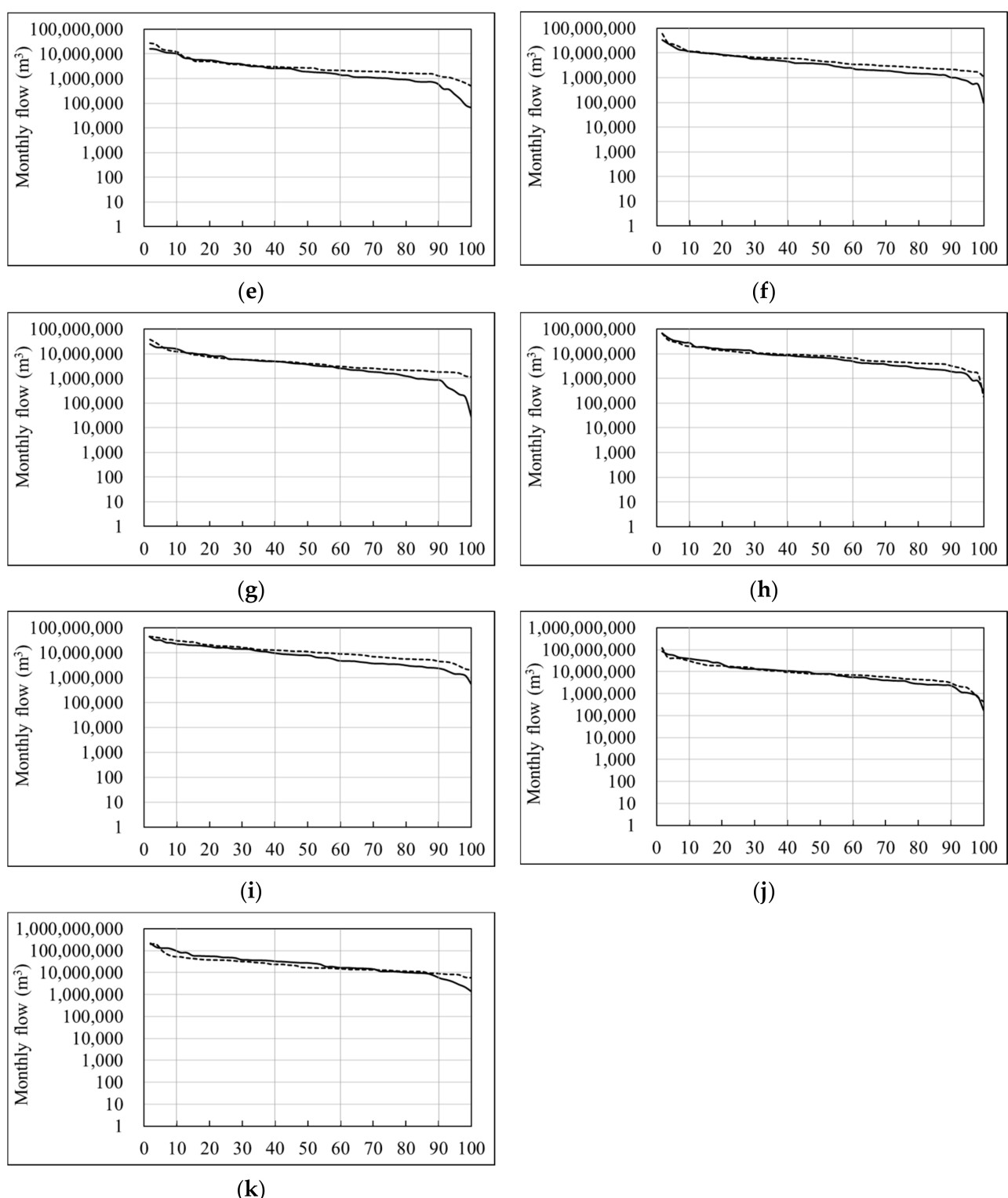

**Figure 4.** Flow duration curves of separated monthly baseflow and estimated monthly baseflow in calibration: (**a**) Wsd-01; (**b**) Wsd-02; (**c**) Wsd-04; (**d**) Wsd-06; (**e**) Wsd-08; (**f**) Wsd-10; (**g**) Wsd-12; (**h**) Wsd-14; (**i**) Wsd-16; (**j**) Wsd-18; (**k**) Wsd-20.

Therefore, the application of optimized coefficients to each watershed will render the estimated monthly baseflow similar to the separated monthly baseflow. Thus, the application of the finally determined coefficients is expected to result in satisfactory predictions.

### 3.2. Validation of Regression Model Coefficients

When the model parameters that were determined for use for calibration were adjusted for the associated watersheds during calibration, the results derived for these watersheds could be considered satisfactory. However, it is necessary to apply these model parameters to watersheds other than those used in the calibration process to examine whether the model parameters were well calibrated and determine whether the estimated results are applicable. Therefore, in this study, the values determined for the coefficients that correspond to the model parameters in the calibration process were applied to the second group to determine whether the estimated monthly baseflow is also applicable in this group.

The estimated monthly baseflow was determined to be applicable because the $R^2$ ranged from 0.618 (Wsd-09) to 0.786 (Wsd-17) and the NSE from 0.567 (Wsd-07) to 0.727 (Wsd-05) (Table 6). Based on the scatter plot, the estimated monthly baseflow tended to be slightly lower than the separated monthly baseflow for Wsd-03, Wsd-11, and Wsd-19. Overall, however, no significant differences were observed in either tendency or value for the estimated and separated monthly baseflow (Figure 5).

**Table 6.** $R^2$ and NSE of separated and estimated monthly baseflow in the second watershed group.

| Watershed | $R^2$ | NSE |
|---|---|---|
| Wsd-03 | 0.630 | 0.569 |
| Wsd-05 | 0.739 | 0.727 |
| Wsd-07 | 0.664 | 0.567 |
| Wsd-09 | 0.618 | 0.606 |
| Wsd-11 | 0.658 | 0.647 |
| Wsd-13 | 0.689 | 0.632 |
| Wsd-15 | 0.728 | 0.670 |
| Wsd-17 | 0.786 | 0.674 |
| Wsd-19 | 0.626 | 0.571 |

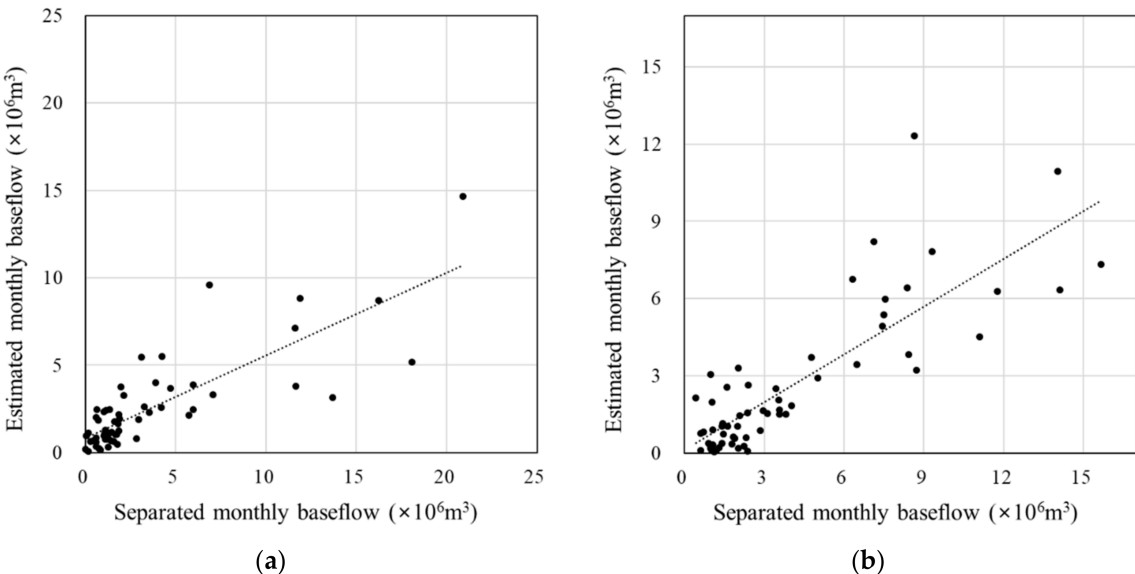

(a)  (b)

**Figure 5.** *Cont.*

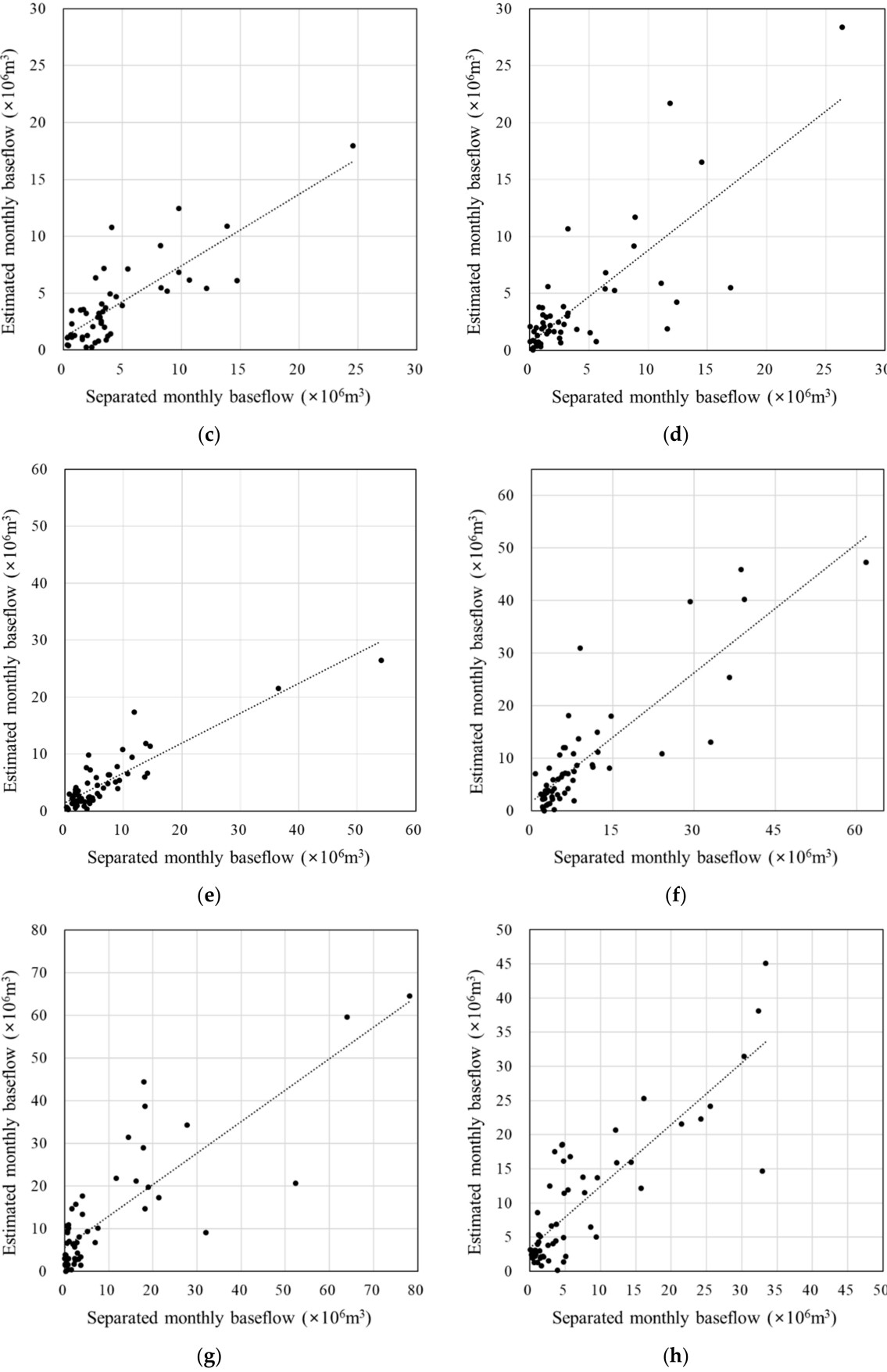

**Figure 5.** *Cont.*

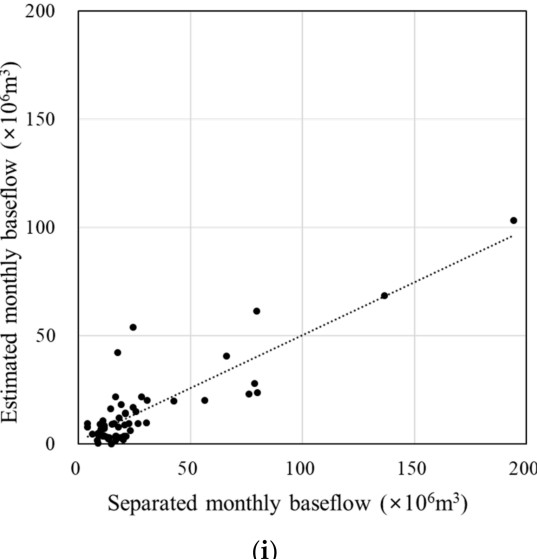

(**i**)

**Figure 5.** Scatter plot of separated monthly baseflow and estimated monthly baseflow: (**a**) Wsd-03; (**b**) Wsd-05; (**c**) Wsd-07; (**d**) Wsd-09; (**e**) Wsd-11; (**f**) Wsd-13; (**g**) Wsd-15; (**h**) Wsd-17; (**i**) Wsd-19.

Similar to the results of calibration, the estimated monthly baseflow did not capture the separated monthly baseflow in the dry-conditions and the low-flow regimes often in the flow duration curve plots; however, it does in the other flow regimes, which are the high-flow, the moist-conditions, and the mid-range flow regimes (Figure 6). Based on flow duration curves in both calibration and validations processes, the estimated monthly baseflow fit to the separated monthly baseflow reasonably in the high-flow and the moist-conditions; however, it did not especially in the low-flow regime. This means that the monthly baseflow approach will be reasonable in the applications with the issues regarding the high flow and the moist conditions such as flooding or nonpoint source pollution analysis. However, caution needs to be exercised when the approach is used for any applications regarding low flow such as water supply simulations in drought.

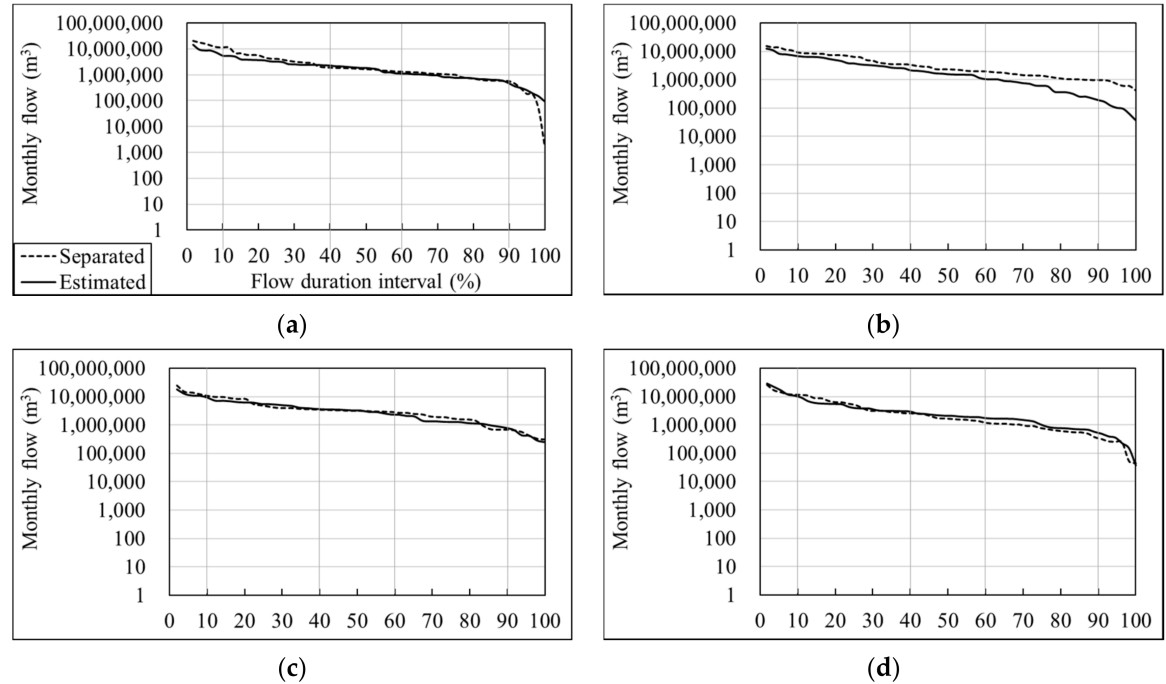

**Figure 6.** *Cont.*

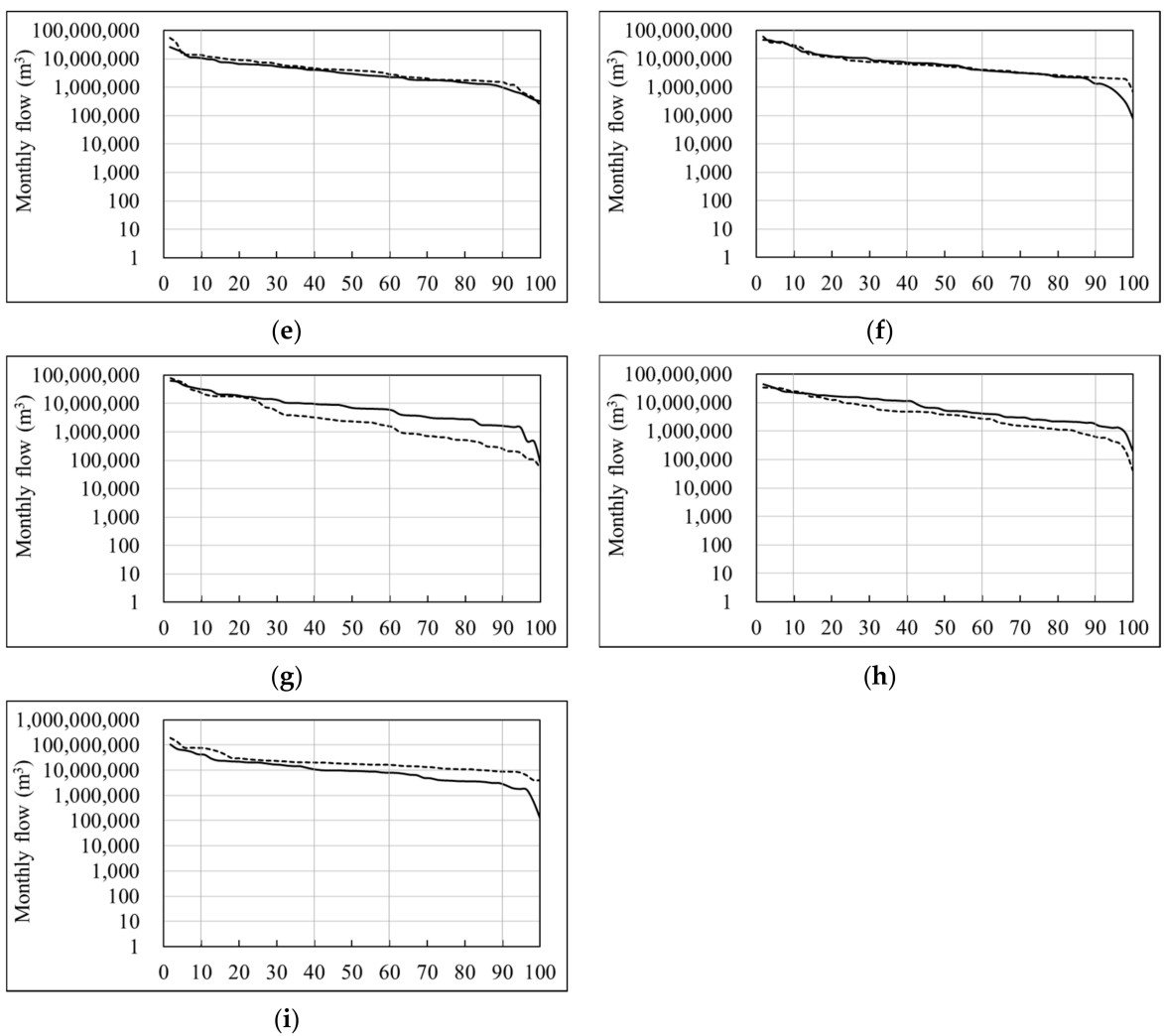

**Figure 6.** Flow duration curves of separated monthly baseflow and estimated monthly baseflow in validation: (**a**) Wsd-03; (**b**) Wsd-05; (**c**) Wsd-07; (**d**) Wsd-09; (**e**) Wsd-11; (**f**) Wsd-13; (**g**) Wsd-15; (**h**) Wsd-17; (**i**) Wsd-19.

## 4. Conclusions

This study aimed to propose a method for predicting the baseflow in order to improve the shortcomings of the current long-term hydrologic impact analysis (L-THIA) model, which can only predict surface runoff. To achieve this aim, the measured streamflow was separated into surface runoff and baseflow to determine whether the L-THIA model can predict the hydrologic behavior in a given watershed through comparison with the measured flow. The model is limited in that it cannot determine the influence of the baseflow because only the surface runoff can be estimated. It has been used until now because it requires only land use maps, soil maps, and precipitation data and its computation process is simple from a user point of view. Therefore, it is necessary to improve the current L-THIA model so that it can predict the baseflow while maintaining its simplicity. As such, a method to improve the limitations of the L-THIA model was proposed by applying the baseflow prediction method based on land use and annual precipitation, which is used in the spreadsheet tool for the estimation of pollutant load (STEPL) model.

To this end, 20 independent watersheds in South Korea were selected and divided into two groups. The first group of 11 watersheds was used to propose the monthly baseflow prediction method, and the second group of nine watersheds was used to examine the proposed method. Since the proposed method uses monthly precipitation data along with the areas and coefficients associated with seven different land use types, it does

not exceed the range of data used in the current L-THIA model to predict surface runoff. The simplicity of the model can, therefore, be maintained. In addition, the coefficients represent the contribution of precipitation to the baseflow for each land use type. In the case of urban areas, which are generally considered to have a low contribution to the baseflow due to their high impervious surface ratio, the coefficient was determined to be lower than that of other land uses. Wetland is considered to have a high contribution to the baseflow because of the continuous infiltration that is associated with water storage. Therefore, it has a higher coefficient than the other land use types. In addition, ponded paddy fields, which are characteristic of the agriculture in Korea, are assumed to contribute to the baseflow in a similar manner as wetland during the cultivation period from May to October. Therefore, the coefficient for agricultural land was determined to be similar to the that for wetland. In other words, it appears that the proposed monthly baseflow prediction method can sufficiently reflect the conditions of the different land uses while reflecting seasonal conditions as it uses monthly precipitation.

A proposed step in use of the approach is to separate direct runoff and baseflow from measured streamflow first, to calibrate direct runoff estimated with curve numbers, and to calibrate baseflow estimated by the suggested method, in turn. Additionally, since the proposed method does not reflect the hydrologic behavior beneath the surface and can be seen as a statistical or probabilistic approach based on the characteristics of the watersheds in Korea, caution needs to be exercised when it is applied to watersheds without ponded paddy fields. Attention also needs to be paid to the application of this monthly baseflow prediction regression equation when the watershed area used in the derivation process is exceeded. In addition, the suggested regression was developed on a monthly basis; therefore, it will be limited to apply for daily or weekly baseflow estimations.

**Author Contributions:** Conceptualization, Y.-S.P.; methodology, H.-S.C.; investigation, M.-S.C.; data curation, M.-S.C.; writing—original draft preparation, H.L.; writing—review and editing, Y.-S.P.; visualization, H.-S.C.; supervision, Y.-S.P. All authors have read and agreed to the published version of the manuscript.

**Funding:** This subject is supported by Korea Ministry of Environment as "The SS (Surface Soil conservation and management) projects; 2019 (2019002820001)".

**Institutional Review Board Statement:** Not applicable.

**Informed Consent Statement:** Not applicable.

**Data Availability Statement:** The data presented and used in the study are available on the request from the corresponding author.

**Acknowledgments:** The authors send special thanks to the Korea Environmental Industry & Technology Institute for continuous research project support.

**Conflicts of Interest:** The authors declare no conflict of interest.

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
