# Peer review of "A Study to Suggest Monthly Baseflow Estimation Approach for the Long-Term Hydrologic Impact Analysis Models: A Case Study in South Korea"

_water, doi:10.3390/w13152043_

Round 1

Reviewer 1 Report

Ms. Ref. No.: water-1261537

Title: A Study to Suggest Monthly Baseflow Estimation Approach for the Long-Term Hydrologic Impact Analysis Models

Journal: Water

General Comments

This is no exception in South Korea, where land-use/land-cover has undergone a series of dramatic changes over the years mainly due to the ever-growing large population. However, estimating the impact of such changes on a wide range of ecosystem services is seldom attempted. The authors have done a good amount of work to justify the assessment of the changes in the hydrologic behavior of the model by using the hydrological model at several catchments. However, there are some points described below that have to be considered before publication. The overall presentation in the Introduction section lacks synergy and exists in bits and pieces. Though authors have identified the research gaps the literature survey part can be more streamlined and while coming towards the problem statement.

Authors have missed discussing the important aspect of incorporation of the effect of landuse/landcover on runoff in different scenarios. There is a vast literature on this I would like to suggest following to this which author should add is “Vegetation can have a significant effect on hydrological fluxes due to variations in the physical characteristics of the land surface, soil, and vegetation; such as the roughness, albedo, infiltration capacity, root depth, architectural resistance, leaf area index (LAI), and stomatal conductance (Srivastava et al., 2020)”.  I would recommend adding these recent references to add more scientific weight in their Introduction.

Srivastava, A., Kumari, N. & Maza, M. (2020). Hydrological Response to Agricultural Land Use Heterogeneity Using Variable Infiltration Capacity Model. Water Resour Manage 34, 3779–3794. https://doi.org/10.1007/s11269-020-02630-4

Aghsaei, H., Dinan, N. M., Moridi, A., Asadolahi, Z., Delavar, M., Fohrer, N., & Wagner, P. D. (2020). Effects of dynamic land use/land cover change on water resources and sediment yield in the Anzali wetland catchment, Gilan, Iran. Science of the Total Environment, 712, 136449.

Please provide the link for the sources from where the data is obtained and also how the cloud cover was handled, is there any threshold that authors have taken to consider the cloud cover and radiometric correction.

I would suggest the authors provide a flowchart describing the preprocessing of raw datasets obtained from different sources followed by obtaining the confusion matrix and post-classification change.

It is hard to find the assessment of classification accuracies such as producer’s accuracy, users’ accuracy, and kappa value. Please add them in the manuscript.

Author Response

Thank you for reviewing the manuscript.

Please find the attached file for our response.

Reviewer 2 Report

  • Line 2: The study tried to predict baseflow in a simplified approach, thus it seems to be a statistical approach. Therefore, the method presented in the study reflects the spatio-temporal characteristics of the Korean watersheds applied to determine the regression model coefficients, and it is required to review or examine them before applying them to other regions. Therefore, I suggest that the article indicates that the study is a case for Korea.
  • Line 145: It is required to revise font and line style for the Wsd-01.
  • Line 146: The study suggests an approach to predict monthly baseflow in Korean watersheds. I wonder if there is any specific reason that it is monthly only, not for daily nor yearly.
  • Line 146: The study suggests an approach to predict monthly baseflow, based on the STEPL model approach that requires not only land uses but also HSGs as well. However, the study does not consider HSGs while the STEPL model does. I wonder if the regression model accuracy could have been better with HSGs.
  • Line 174: The basic formula of the regression looks somewhat complicated. It is strongly suggested to revise it so that it look simpler.

Author Response

(The authors gave the same response as above.)

Reviewer 3 Report

This work proposed a monthly baseflow estimation approach in the regions of interest with the long-term hydrologic impact analysis model in South Korea. As learned from the manuscript, it is well written and organized. It also satisfied the scope the Water Journal. I would like to recommend a minor revision before publication as there are a few issues to be addressed:

  1. The monthly baseflow estimation approach is compared with the baseflow separated monthly baseflow. As some bias might exist in the baseflow separation method. It is difficult to claim the robust of the proposed baseflow estimation approach. I would like to suggest the authors use some observation of baseflow data to validate this proposed approach in the revision.
  2. In the discussion, it is necessary to address the potential capability in the weekly or daily baseflow estimation.

Author Response

(The authors gave the same response as above.)

Reviewer 4 Report

The manuscript "A Study to Suggest Monthly Baseflow Estimation Approach for the Long-Term Hydrologic Impact Analysis Models" of Lee et al. is a technical report, which describes the application of the Long-Term Hydrologic Impact Analysis (L-THIA) model to estimate long term changes of monthly baseflow due to land use change.

In general land use change an its implications on the hydrology, especially discharge, is relevant to the hydrological community. Although the manuscript is intended to be a  technical report, there is a major deficiency since comparisons to other existing studies with other models are missing. At the very beginning the authors should mention other modelling studies on land use change like https://doi.org/10.1007/s10113-013-0429-3 and https://doi.org/10.1007/s40710-015-0099-x. In the following introduction, there is a strong focus on the L-THIA, but it would be necessary give at least some comparisons to other models and studies.

The weakest point of the manuscript is the evaluation of model performance and its visualisation. There is no presentation of baseflow dynamics although it is evaluated with NSE. The current plots are not sufficient to decide about the model performance since deviations at the low flow can not be inspected. Furthermore, R2 is in this case misleading since deviations wihtin the high flow are overweighted. This is also the case using the NSE. Therefore, it would be a good option to extract additional information using the flow duration curve (e.g. https://doi.org/10.1111/j.1752-1688.1995.tb03419.x). Examples of how to use different segments of the flow duration curve or the flow duration curve itself are https://doi.org/10.1002/hyp.11300, https://doi.org/10.5194/hess-15-2805-2011, https://doi.org/10.1016/j.jhydrol.2013.12.044.

Before the manuscript and its results can be evaluated, these points need to be revised.

Author Response

(The authors gave the same response as above.)

Round 2

Reviewer 1 Report

This paper is very interesting for the hydrologic community. It is well written. In my opinion, this is an interesting and valuable contribution to hydrologic science. After thoroughly going through the revised manuscript, authors have addressed all of the comments and in doing so they have improved the quality of the manuscript.  The authors have addressed all previous concerns expressed by the reviewers and in the process have improved the work, confirmed the validity of their findings and gained confidence in their introduction, methods, results and conclusions. I would like to congratulate the authors for an interesting and well executed work and I recommend this manuscript for publication in Water (MDPI) in its current form.

Author Response

Thank you for your comment.

Again, I would like to say that I appreciate for your kind comments in the previous review.

Especially the comment regarding a schematic flow plot of the study will be helpful for the readers.

I hope you have a great day today!!

Reviewer 4 Report

The manuscript "A Study to Suggest Monthly Baseflow Estimation Approach for the Long-Term Hydrologic Impact Analysis Models: A Case 
Study in South-Korea" of Lee et al. is a revised version of a technical report.

According to the review of the previous manuscript, I still see shortcomings in model evaluation. As already explained in the first review, I am still not convinced using NSE and R2 as evaluation criteria. The authors argue that the flow duration curve is not appropriate in this study example. However, there are several studies published, showing of how to use the FDC for model evaluation for discharge but also other hydrological components at different spatial and temporal scales. The authors argue, that the FDC concept is not applicable for their approach with respect to considered tempotal resolution. However, if this argument would be true, I ask the authors to justify the use of NSE in their study since NSE is a criterion to evalutate the goodness of fit for dynamics in data-time-series. In this regard, I also question why the authors make use of the NSE but do not show any time series within their study. If the NSE is appropriate to evaluate the simulated data of the study, then the FDC would be too. FDC can be constructed from hourly, daily, monthly and yearly hydrological data. These FDC can be used for comparison of measured and simulated data. I guess that calculating FDC might be time consuming so that the authors try to avoid this piece of extra work. However, in my opinion the study and its results would be more convincing. Without some additional model evalution, I cannot retrace the results and conclusions.

Author Response

Thank you for the comment.

As you suggested again to provide flow duration curves, we added them.

Please find the edits that are figures 4 and 6.

I hope you have a great day today.

Thanks.

Round 3

Reviewer 4 Report

The current manuscript "A Study to Suggest Monthly Baseflow Estimation Approach for the Long-Term Hydrologic Impact Analysis Models" by Lee et al. is the second revision of previous manuscripts. During the review process several shortcomings were identified, which were improved partly in the current manuscript. As already mentioned in the first review, I still see some potential to improve the soundness of the study. The authors included the evaluation of their results by the flow duration curve to a limited extent. As already recommended in the first round of the review, I still see the need to refer to former studies: 

"Therefore, it would be a good option to extract
additional information using the flow duration curve (e.g. https://doi.org/10.1111/j.1752-
1688.1995.tb03419.x). Examples of how to use different segments of the flow duration curve or the
flow duration curve itself are https://doi.org/10.1002/hyp.11300, https://doi.org/10.5194/hess-15-2805-
2011, https://doi.org/10.1016/j.jhydrol.2013.12.044."

The authors show some FDC in the current manuscript, but there are obviously some deviations between observed and simulated data. An explanation for this deviation is not discussed properly an there is the need to quantify the model performance in these segments of the flow duration curve.

Since this point was addressed by the reviewer two times, it is to the editor to decide about publication of the study.